# Cognitive Decline and Its Associated Factors in Patients with Major Depressive Disorder

**DOI:** 10.3390/healthcare11070950

**Published:** 2023-03-25

**Authors:** Husni Zaim Ab Latiff, Suthahar Ariaratnam, Norley Shuib, Mohamad Rodi Isa

**Affiliations:** 1Department of Psychiatry, Faculty of Medicine, Universiti Teknologi MARA (UiTM), Cawangan Selangor, Kampus Sungai Buloh, Sungai Buloh 47000, Selangor, Malaysia; 2Psychiatric and Mental Health Department, Hospital Sultanah Nurzahirah, Kuala Terengganu 20400, Terengganu, Malaysia; 3Department of Public Health Medicine, Faculty of Medicine, Universiti Teknologi MARA (UiTM), Cawangan Selangor, Kampus Sungai Buloh, Sungai Buloh 47000, Selangor, Malaysia

**Keywords:** cognitive decline, major depressive disorder, outpatients, Montreal Cognitive Assessment, Mini International Neuropsychiatric Interview

## Abstract

Background: Major Depressive Disorder (MDD) is a significant and common mental health problem occurring worldwide. Cognitive decline is frequently observed during acute and residual phases of MDD, contributing significantly to functional impairment. The aim of this study was to determine the clinical profile and correlates of cognitive decline amongst adult outpatients with MDD. Methods: The survey was cross-sectional in design. A systematic random sampling method was used to recruit patients. Confirmation of MDD was achieved by using the Mini International Neuropsychiatric Interview (M.I.N.I 7.0). Cognitive decline was measured using the Montreal Cognitive Assessment (MoCA). Descriptive analysis was performed, followed by univariate and multiple logistic regression analyses. Results: Out of 245 patients, 32.7% (*n* = 80, 95% CI: 26.7, 38.6) had cognitive decline. Multiple logistic regression showed the existence of cognitive decline amongst MDD patients, which was significantly associated with those having secondary and lower levels of education (OR: 6.09; 95% CI: 2.82, 13.16; *p* < 0.001), five or more depressive episodes (OR: 8.93; 95% CI: 3.24, 24.67; *p* < 0.001), treatment non-compliance (OR: 3.48; 95% CI: 1.40, 6.59; *p* = 0.003), and medical comorbidity (OR: 2.74; 95% CI: 1.46, 5.18; *p* = 0.002). Conclusions: Cognitive decline is a prevalent condition among outpatients with MDD. Clinicians need to be cognizant about measures of cognition and related risk factors. Timely control of both depression and medical comorbidities would be a reasonable approach to improve functional outcomes in MDD patients.

## 1. Introduction

Major Depressive Disorder (MDD) is a common mental health problem that disrupts a person’s mood and adversely affects one’s psychosocial as well as occupational functioning.

Cognitive decline usually refers to deficits in attention, verbal and nonverbal learning, short-term and working memory, visual and auditory processing, problem-solving, processing speed, and motor functioning [1]. An array of cognitive decline has been frequently observed in patients suffering from MDD [2], which has regrettably persisted beyond remission [3,4,5]. In addition, systematic reviews and meta-analyses involving patients with major depressive episode (MDE) and healthy controls found that cognitive decline remained, despite achieving remission among those with MDE [6,7]. However, this association with depressive symptoms remains relatively unknown [2], including its clinical relevance, in particular whether a specific cognitive domain could relate to biological phenotypes that in turn could impact the choices of anti-depressant and/or non-pharmacological interventions [8].

The prevalence of cognitive decline among unmedicated patients with MDD as measured by the Measurement and Treatment Research to Improve Cognition Schizophrenia Consensus Cognitive Battery (MCCB) was 23.13% [9]. Indeed, cognitive symptoms were present in 85–94% and 39–44% of the time during depressive episodes and remissions in patients with MDD, respectively [5]. Significant executive function deficits were observed in about 20–30% of patients with MDD [10]. Additionally, the Study on Aspects of Asian Depression (SAAD), which was conducted in six Asian countries (China, Korea, Malaysia, Singapore, Taiwan, and Thailand), found that 67.4% and 73.2% of medication-free, non-elderly Asians with MDD had subjective memory deficit and subjective cognitive decline, respectively [11]. A subgroup analysis of 211 Malaysian patients recruited from the Cognitive Dysfunction in Asian Patients with Depression (CogDAD) study has shown that clinically relevant cognitive decline was reported by 54.5% of patients [12].

Cognitive function is a domain of MDD symptomatology that is severely affected in older patients [13,14]. Those with higher education and occupational attainment were able to endure greater brain pathology before demonstrating functional decline [15]. Cognitive decline was postulated as a sequalae or residual feature of an acute MDE. This postulation was supported by objective findings from a study that found that the number of depressive episodes and duration of illness were significantly associated with cognitive decline [16]. Those with early-onset depression were found to have better performance in memory and verbal learning domains when compared to patients with late-onset depression (first depressive episode after 65 years of age) [17]. The phase of illness was also found to be an associated factor of cognitive decline in patients with MDD. In a 3-year follow-up study, 94% of patients with MDD reported cognitive symptoms during the acute phase of the illness, while 44% still experienced similar complaints despite full or partial remission of depressive symptoms while on treatment [5]. Antidepressants (ADs) have a positive effect on cognitive function [18]. Nevertheless, there is recent evidence to support the superiority of Vortioxetine as a better pro-cognitive effect drug than other older-generation ADs [19,20,21].

Literature pertaining to critical analysis of correlates between cognition decline with other common mental disorders such as schizophrenia and bipolar disorder has been rather robust in comparison to MDD patients [10]. Apart from the study by Zainal et al. [12] using the Perceived Deficits Questionnaire (PDQ-D) to assess cognitive decline, there were a limited number of studies published locally that have specifically addressed the issue of cognitive decline in patients with MDD. Moreover, during daily clinical practice, psychiatrists usually assessed the patient’s subjective sense of cognitive function without using appropriate clinical tools [22].

Given the above considerations, the present study was performed to determine the prevalence of cognitive decline and the different cognitive domains affected using the Montreal Cognitive Assessment (MoCA) [23] tool amongst adult outpatients with MDD during a routine clinical practice. In addition, we studied the association between cognitive decline and age, gender, ethnicity, marital status, education level, employment status, household income, duration of illness, age at the onset of illness, symptomatic status, number of episodes, types of medication, compliance of medication, substance usage, and medical comorbidities to analyze the possible links affecting cognitive function among MDD outpatients.

## 2. Methods

### 2.1. Study Population

This was a cross-sectional study conducted at the Department of Psychiatric and Mental Health, Selayang Hospital, Selangor, Malaysia. Specifically, the general adult outpatient clinic was utilized to recruit patients. The clinic operates every Monday and Wednesday. Hence, all general adult patients with mental health disorders, including those with depression and bipolar disorder, were seen in this clinic.

Patients had to fulfil the following inclusion criteria: (1) outpatient; (2) aged 18–65 years old; (3) MDD established after assessment with the Mini International Neuropsychiatric Interview; (4) provided informed consent. Patients were excluded if they had the following conditions: (1) new case or first psychiatric contact; (2) women who were pregnant/breastfeeding or in the 4 weeks following delivery; (3) concomitantly partaking in another clinical trial; (4) acutely suicidal or psychotic; (5) other psychiatric comorbidities such as bipolar depression, anxiety disorder, schizophrenia, obsessive compulsive disorder, persistent depressive disorder, eating disorders, substance use and addictive disorder, and neurocognitive disorder; (6) undergoing ECT.

After fulfilling the inclusion and exclusion criteria, patients were recruited based on clinical diagnosis of major depressive disorder. Subsequently, MINI was performed to confirm the diagnosis.

Sample size was determined based on a study that affirmed the prevalence of cognitive dysfunction in MDD patients as 20% [10]. Subsequently, the sample size was decided by using OpenEpi Sample Size Calculation Software with the following formula: 5% significant level (Z_1−α_ = 1.96), prevalence of 20% (*p* = 0.2), absolute precision (D) of 5% and estimated patients attending psychiatric clinic, Hospital Selayang as 2000, the sample required was 220. After adjusting for a 10% dropout rate, the minimum sample required was 244.

Systematic random sampling (SRS) was applied in this study. SRS is one of the probability sampling methods whereby every respondent has an equal chance to be chosen from a target population. It was applied in this study using computer-generated numbers. The registered patients were given an identified number. The patients with the generated number were selected for the study. The participants were selected according to a random starting point and a fixed-periodic interval. Fixed-periodic interval, also known as the sampling interval, is calculated by dividing the population size by the sample size. The first respondent was selected randomly from the numbers 1 to 9. Then, the next selected respondent was selected using the fixed-periodic interval and so on until the desired sample size was achieved.

The patients who fulfilled the selection criteria were approached during their clinic visits. They were briefed on the background and objectives of the study and were given the participant information sheet. Informed verbal and written consents were obtained prior to data collection. Those who consented to participate were asked to provide their sociodemographic data (age, gender, ethnicity, marital status, employment status, educational level, employment status, and household income). The clinical profile was data pertaining to the patients’ clinical information (duration of illness, age at the onset of illness, symptomatic status, number of episodes, types of medication, perceived current compliance, lifetime substance usage, recent substance usage, and medical comorbidity).

### 2.2. Study Assessments

Next, patients were interviewed using Mini International Neuropsychiatric Inventory (M.I.N.I 7.0) to confirm the diagnosis of MDD. M.I.N.I was developed as a reliable and acceptable diagnostic tool for psychiatric disorders in a clinical setting and for research purposes [24]. M.I.N.I 7.0 is a short, structured diagnostic interview for psychiatric evaluation and assesses the 17 most common disorders in mental health. Module A, which consisted of 6 items (A1–A6), was used to confirm the diagnosis of MDD. The diagnosis of MDD was made when patient had answered “Yes” for 5 or more in the items A1–A3 and answered “Yes” for item A4. Item A1 explores persistent low mood while item A2 is related to anhedonia. Item A3 explores depressive symptoms according to DSM-5 criteria. Item A4 denotes significant impairment of functioning level due to depressive symptoms. This tool was also used to identify if the patient was having a current major depressive episode (MDE), or had a past episode of MDE (in remission), or having recurrent episodes of MDE. A current episode of MDE was coded when the patient fulfilled the MDE criteria in the past 2 weeks and the reverse for a past episode of MDE. Recurrent MDE was coded if the patient did not experience significant depressed mood or anhedonia for an interval of at least 2 months in between 2 episodes.

The patients were then screened for cognitive decline using the Montreal Cognitive Assessment (MoCA) tool, which is a brief cognitive screening test with high sensitivity and specificity to detect cognitive impairment [25]. The MoCA test has 30 questions that help to assess how an individual’s cognitive function is affected. It evaluates different types of cognitive abilities such as orientation, short-term memory, executive function, language abilities, abstraction, animal naming, attention, and the clock-drawing test. The scores on the MoCA ranged from zero to 30 with a score of 26 and higher generally considered normal. A total score below 26 indicates cognitive decline. Patients were administered the Montreal Cognitive Assessment (MoCA) tool in their preferred language (either English or Malay). The Malay version of MoCA was a validated tool for brief cognitive screening with high sensitivity and specificity [26]. Prior to the study, permission to use both versions of MoCA was obtained and granted by the respective authors. Although the instrument is a screening measure for MCI and Alzheimer’s disease, MoCA can be useful for the evaluation of cognitive functioning in depressed patients [27]. This instrument, which has already been translated into several languages, including the Malay language, is a validated tool that requires relatively brief cognitive screening compared to a complete set of neurocognitive tests. There is also another study that supports the use of this tool for screening of neurocognitive deficits in depressed patients [28]. The study has shown that MoCA and its subtests are reliable as well as valid for objective assessment of cognitive performance in patients with MDD and could be a tool for screening neurocognitive deficits in depressed patients.

### 2.3. Ethical Approval

Approval for this study was obtained from the Research Ethics Committee of Universiti Teknologi MARA (UiTM). Additionally, ethics approval from the Medical Research and Ethics Committee (MREC) of the Ministry of Health was obtained via the National Medical Research Registry (NMRR) (Protocol number NMRR-19-702-46518).

### 2.4. Statistical Analyses

The data collected were entered and analyzed using SPSS version 25.0. The descriptive statistics were presented depending on the types of variables. For the continuous variable, the data were presented based on the type of normality. When the data were normally distributed, it was presented by mean and standard deviation. When the data were not normally distributed, it was presented using median and interquartile range (IQR). For the categorical variable, data were presented by frequency and percentage.

For the categorical variable, the comparison was analyzed using chi-square test. In terms of continuous variable, the comparison was analyzed using independent *t*-test. The determination of the affected cognitive domain in the study population was presented descriptively. Factors associated with cognitive decline were analyzed in two steps. The first step was to calculate the crude odds ratio (OR) and to select the significant factor associated with cognitive decline using univariate analysis (simple logistic regression). Any significant factors were then analyzed using multiple logistic regression (MLogR) to calculate the adjusted odds ratio (Adj. OR) and to adjust for the confounding factors. All the assumptions were checked, including the model fit, multicollinearity, sensitivity, and specificity. The Receiver Operative Characteristic (ROC) curve was developed to determine how the statistical model discriminates between patients with cognitive decline and non-cognitive decline. A *p*-value of <0.05 was considered statistically significant.

## 3. Results

A total of 245 patients participated in the study. Table 1 summarizes the sociodemographic data of the patients. The mean age of the patients was 42.7 ± 14.7 years. The majority of them were in the age group of 56 to 65 years (28.6%), females (64.9%), Malays (46.5%), married (51%), receiving secondary education (45.7%), and unemployed/being a housewife (36.3%). More than two-thirds (81.2%) of the household income originated from the group of Bottom 40 (who have earnings of less than RM 3860).

Table 2 illustrates the clinical profile of the patients. About half of the patients were diagnosed with MDD for a duration of 1–5 years (51%). The onset of MDD was observed to be the highest in the age group of 26–35 years old (24.1%). In terms of symptomatic status, 6.5% of the patients had a current episode of MDD. About half of the patients had experienced 3–4 episodes of relapse (51%) and 25.7% had 5 or more episodes of relapse in the past. The majority of the patients were treated with typical antidepressants, which consisted of selective serotonin reuptake inhibitor (SSRIs) or serotonin-noradrenaline reuptake inhibitors (SNRIs) (68.2%) and reported good compliance to their medication (79.7%). Regarding substance usage, the majority did not have a history of substance use, either lifetime intake (84.9%) or recent consumption in the past week (97.6%). Medical comorbidities were observed in 42.1% of the patients, of which 22.9% had multiple medical conditions.

The mean total score of the MoCA assessment among the study populations was 26.56 (±3.13). Cognitive decline was observed in 80 patients (32.7%). Hence, the prevalence of cognitive decline among MDD patients was 32.7% (95% CI: 26.7, 38.6). Among 80 MDD patients with cognitive decline, the highest frequency of affected cognitive domain (refer Table 3) was attention (31; 38.7%), followed by memory (22; 27.5%) and visuospatial/executive function (18; 22.5%).

The final model was adjusted for age group, ethnicity, education level, employment status, duration of illness, symptomatic status, number of episodes, compliance to medication, and medical comorbidities. Multicollinearity and interaction were checked and revealed no issues. Hosmer Lemeshow test *p* value = 0.879, Classification Table (overall correctly classified percentage = 74.2%) and ROC curve (area under ROC curve = 79.8%) were accepted to check model fitness.

Cox & Snell R^2^ = 0.245; B = beta; SE = standard error; df = degree of freedom; OR = odds ratio; CI = confidence interval.

Following MLogR analysis (refer Table 4), four factors remained statistically significant, which were education level, number of episodes, compliance to medication, and medical comorbidities. MDD patients who had a secondary education level or lower were 6.09 times more likely to have cognitive decline when compared to individuals with an education level higher than secondary level. Patients with 5 depressive episodes or more were 8.93 times more likely to have cognitive decline as compared to those having 1–2 episodes of depression. MDD patients who did not take the medication were 3.5 times more likely to experience cognitive decline when compared to patients who had good compliance to medication. For medical comorbidity, MDD patients were 2.7 times more likely to have cognitive decline when compared to patients without medical comorbidity.

## 4. Discussion

This study indicates that the prevalence of cognitive decline amongst adult outpatients with MDD was 32.7%. The prevalence was higher compared to a previous study involving outpatients aged from 18 to 65 years old with MDD, which recorded a prevalence of 11.8% at 1.0 standard deviation (SD), 2.9% at 1.5 SD, and 1.5% at 2.0 SD cut-offs on a global cognitive composite [29]. The lower prevalence can be explained by the different assessment tools used for cognitive decline, whereby the previous study used multiple neurocognitive assessments to evaluate various cognitive domains.

From our study, the most affected cognitive domains were attention, memory, and visuospatial/executive function, which could be contributed by symptoms of MDD itself, such as difficulty in concentration and psychomotor retardation. This finding was comparable to a study comprising a total of 165 patients, which found that depressed outpatients had significant deficits in the domains of selective attention and working memory [30]. In addition, depressive mood significantly predicted verbal memory performance for short-term recall, long-term recall, and recognition memory among elderly patients receiving less than 6 years of formal education [31].

The hippocampus has a vital role in memory. An apparent neuropathological feature of depression is reduction in hippocampal volume, which is seen in both MRI [32] and post-mortem [33] studies. A high cortisol level in depression is toxic to the hippocampus, which leads to its decline [34]. Dysregulation of the dorsal and lateral prefrontal cortex while performing executive tasks has been observed in depressed patients in a functional neuroimaging study [35].

Pertaining to education level, patients with secondary or lower levels of education were six times more likely to have cognitive decline in MDD individuals than those with an education level higher than secondary level. This finding concurred with other reports [10,30,31] that had examined cognitive performance among participants with MDD and discovered poorer overall cognitive function was associated with lower education level. In other words, higher educational attainment provided better protection against cognitive decline. This observation could possibly be explained by the concept of cognitive reserve. Cognitive reserve refers to the ability to make flexible and efficient use of cognitive networks when performing tasks in the presence of brain pathology [36]. Cognitive reserve provides a ready explanation for why many studies have demonstrated that higher levels of intelligence as well as educational and occupational attainment are good predictors of which individuals can sustain greater brain damage before demonstrating functional deficit [15].

With regard to number of depressive episodes and cognitive decline, this outcome conformed to other studies that have revealed that having more episodes of MDD was significantly associated with cognitive decline. For example, in a large sample study of 8229 outpatients with MDD, declarative memory decreased approximately 2–3% with each episode of MDD [16]. Additionally, a separate study showed that the existence of memory deficits was a sequela of hippocampus dysfunction and reduction in size of bilateral hippocampus following multiple episodes of MDD [37]. Similarly, greater hippocampal atrophy and—as a possible consequence—deficits in the cognitive domain of memory function have been associated with more episodes of MDD [38]. This has indeed important clinical implications, since hippocampus dysfunction is more apparent during the early years of MDD. Thus, this necessitates early treatment to reduce or eliminate the toxic effect of MDD on brain function that was associated with multiple episodes and chronicity of the MDD [16,37]. Furthermore, current evidence suggests that—to a certain extent—cognitive decline and possibly hippocampal dysfunction are reversible conditions with anti-depressant medications [37].

In terms of compliance to medication, MDD patients who did not take the medication were 3.5 times more likely to endure cognitive decline when compared to individuals with good compliance. Research has consistently demonstrated this effect. Higher risk of recurrence, poor quality of life, poor cognition, and psychosocial functioning were significantly associated with treatment non-compliance [39].

The present study discovered a significant association between medical comorbidity and cognitive decline in MDD patients, with a 2.7-times more likely occurrence of cognitive decline among them compared to patients without medical comorbidity. A case control study affirmed such a finding. With a sample size of 74 participants, that study found that MDD patients with medical illnesses had more significant cognitive decline than depressed controls without medical comorbidity [40]. In addition, it was possible that medical illnesses could compound cognitive decline that was associated with MDD [41]. Therefore, it is only prudent to emphasize concurrent treatment for both MDD and medical comorbidities to alleviate cognitive decline more successfully.

Our study had limitations. Firstly, as it was a cross-sectional survey, we could merely observe cognitive decline prevalence amongst MDD patients in the population during the study period and were unable to infer the causation of these trends. Secondly, most of the participants were in the age group of more than 46 years old. This age group was likely to be marked with subtle cognitive decline. Hence, this could have an impact on the results of this study. Thirdly, this study did not investigate confounding factors such as duration of treatment, nutritional deficiencies or thyroid status, cultural and situational qualities, history of electroconvulsive therapy (ECT), and the presence of comorbid psychiatric disorder (e.g., anxiety disorders or attention deficit/hyperactivity disorder [ADHD]), which could further impair cognitive function in patients with MDD. Fourthly, a more protracted semi-interview such as Composite International Diagnostic Interviews or Structured Clinical Interviews would have been essential, since the use of MINI—which is a brief instrument—tends to yield false-positive results. Lastly, despite the consistent cognitive domains of memory (learning and remembering), attention, and executive function that are commonly affected in patients with MDD, MoCA does not have a clear-cut subscale for thorough assessment of executive function, except for verbal fluency. Hence, a more comprehensive tool is suggested for future studies. Despite these limitations, to the best of our knowledge the present study is among the first that has examined cognitive decline and its associated risk factors amongst outpatients with MDD in Malaysia using the MoCA questionnaire and a systematic random sampling method to select participants. This randomization could reduce selection bias. Hence, a more representative sample was obtained.

## 5. Conclusions

This study could certainly assist physicians and psychiatrists in identifying risk factors associated with cognitive decline so that a more finely tuned treatment plan can be initiated. This entails not only early recognition as well as evaluation of cognition and prompt treatment with anti-depressant medication, but also adequate control of medical comorbidities for improved functional outcomes in MDD patients [42]. Cognitive assessment should be considered during clinical practice as among the important treatment targets for MDD patients.

## Figures and Tables

**Table 1 healthcare-11-00950-t001:** Sociodemographic profile of the patients (*n* = 245).

Variables	Means (SD)	Frequency	
		Number (*n*)	Percentage (%)
Age (years)	42.7 (±14.7)		
Age Group			
18–25		38	15.5
26–35		42	17.1
36–45		42	17.1
46–55		53	21.6
56–65		70	28.6
Gender			
Male		86	35.1
Female		159	64.9
Ethnicity			
Malay		114	46.5
Chinese		94	38.4
Indian		35	14.3
Others		2	0.8
Marital status			
Single, never married		84	4.3
Married		125	51.0
Separated/Divorced		20	8.2
Widowed		16	6.5
Education level			
No formal and Primary education		43	17.5
Secondary education		112	45.7
Tertiary		90	36.7
Employment status			
Unemployed/Housewife		89	36.3
Government sector		25	10.2
Private sector		82	33.5
Self-employed		26	10.6
Student		23	9.4
Household income (RM)			
Bottom 40 (<RM3860)		199	81.2
Middle 40 (RM3860–RM8319)		45	18.4
Top 20 (>RM8319)		1	0.4

**Table 2 healthcare-11-00950-t002:** Clinical profile of the patients (*n* = 245).

Variables	Frequency	
	Number (*n*)	Percentage (%)
Duration of illness		
Less than 1 year	20	8.2
1–5 years	125	51.0
6–10 years	59	24.1
More than 10 years	41	16.7
Age at the onset of illness		
Less than 18 years old	18	7.3
18–25	36	14.7
26–35	59	24.1
36–45	53	21.6
46–55	56	22.9
56–65	23	9.4
Symptomatic status		
Current	16	6.5
Past	155	63.3
Recurrent	74	30.2
Number of episodes		
1–2 times	57	23.3
3–4 times	125	51.0
5 times or more	63	25.7
Types of medication		
Typical antidepressant	167	68.2
Atypical antidepressant	14	5.7
Combination of any psychotropic medications	59	24.1
Not on oral medication	5	2.0
Compliance to medication		
Not at all	5	2.0
Poor compliance	40	16.3
Good compliance	195	79.7
Not on oral medication	5	2.0
Lifetime substance usage		
Yes	37	15.1
No	208	84.9
Recent substance usage in the past week		
Yes	6	2.4
No	239	97.6
Medical comorbidity		
Hypertension	7	2.9
Diabetes Mellitus	10	4.1
Hyperlipidaemia	4	1.6
Stroke	2	0.8
Multiple medical comorbidities	56	22.9
None	142	58.0
Others	24	9.8

**Table 3 healthcare-11-00950-t003:** Descriptive Analysis of Cognitive Domains (*n* = 80).

Variables	Frequency	
	Number (*n*)	Percentage (%)
Most-affected Cognitive Domain		
Attention	31	38.7
Memory	22	27.5
Visuospatial/executive function	18	22.5
Language	3	3.7
Abstraction	3	3.7
Naming	2	2.5
Orientation	1	1.2

**Table 4 healthcare-11-00950-t004:** Multiple Logistic Regression Analysis with Cognitive Decline as Dependent Variable.

	B	SE	Wald (df)	*p*-Value	Adjusted OR (95% CI)
Education level ≤Secondary vs. >Secondary	1.81	0.39	21.174 (1)	<0.001	6.09 (2.82, 13.16)
Number of episodes					
1–2 times			18.024 (2)	<0.001	
3–4 times	1.38	0.48	8.198 (1)	0.004	3.96 (1.54, 10.15)
5 episodes or more	2.19	0.52	17.851 (1)	<0.001	8.93 (3.24, 24.67)
Compliance to medication					
Good compliance			8.962 (1)	0.011	0.995 (0.09, 10.02)
Poor	1.36	0.48	4.345 (1)	0.997	3.48 (1.63, 7.90)
Not at all	2.15	0.52	8.910 (1)	0.003	
Medical comorbidities Yes vs. No	1.01	0.32	9.770 (1)	0.002	2.74 (1.46, 5.18)

## Data Availability

Data is unavailable due to privacy or ethical restrictions. However, they can be made available upon formal request via the Medical Research and Ethics Committee (MREC) of the Ministry of Health, Malaysia.

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
