# Peer review of "Cognitive Decline and Its Associated Factors in Patients with Major Depressive Disorder"

_healthcare, 2023, doi:10.3390/healthcare11070950_

Round 1
Reviewer 1 Report
The study explored the" Cognitive Dysfunction and its Associated Factors in Patients 2 with Major Depressive Disorder". I feel that the study has significant problems and needs to be amended and reconsidered. The biggest problem about this study is that there are a lot of studies already done in this field using stronger methodologies and sample size. for example:
1- The Neuroscience of Depression. Chapter 36 - Determining the cognitive performance in the first episode of depression.Genetics, Cell Biology, Neurology, Behaviour, and Diet. 2021, Pages 389-396
2-Airaksinen E, Larsson M, Lundberg I, Forsell Y. Cognitive functions in depressive disorders: evidence from a population-based study. Psychol Med. 2004 Jan;34(1):83-91. doi: 10.1017/s0033291703008559. PMID: 14971629.
3- Ahern E, & Semkovska M (2017). Cognitive functioning in the first-episode of major depressive disorder: A systematic review and meta-analysis. Neuropsychology, 31(1), 52–72. 10.1037/
neu0000319
4- Depression and Episodic Memory Across the Adult Lifespan: A
Meta-Analytic Review Taylor A. James1,2, Samuel Weiss-Cowie1, Zachary Hopton1, Paul Verhaeghen1, Vonetta M. Dotson3, Audrey Duarte. Psychol Bull. 2021 November ; 147(11): 1184–1214. doi:10.1037/bul0000344
So, what is the merits of this study compared to the previous similar studies?
Another thing is that your tool for measuring cognitive function which is your primary outcome is not good.
One another thing is that (Page 2, lines 87-92 and abstract) It seems you have sample size mistake. while you have calculated your sample size 244, but you included only 80 participants with MDD.
minor mistakes:
page 1, line 41: Your reference number 3 is for animals.
Author Response
|
No |
Comment |
Response |
|
1 |
The study explored the" Cognitive Dysfunction and its Associated Factors in Patients 2 with Major Depressive Disorder". I feel that the study has significant problems and needs to be amended and reconsidered. The biggest problem about this study is that there are a lot of studies already done in this field using stronger methodologies and sample size. for example: The Neuroscience of Depression. Chapter 36 - Determining the cognitive performance in the first episode of depression. Genetics, Cell Biology, Neurology, Behaviour, and Diet. 2021, Pages 389-396 So, what is the merits of this study compared to the previous similar studies? |
Thank you for the comment. Most of the studies of this nature (including the ones mentioned) originated from the global north countries. Conversely but fortunately, this study emerged from a global south population i.e. Malaysia. Both regions are significantly different pertaining to socio-economic and political profiles. Hence, the outcome of this survey on this subject matter especially from the perspective of global south locality would certainly add to the existing data. We hope that with the above explanation, it would meet with your kind approval. |
|
2 |
Another thing is that your tool for measuring cognitive function which is your primary outcome is not good. |
Thank you for the response. We have acknowledged this as one of the limitations for our study under the Discussion section. Accordingly, this statement has been amended to read as “Lastly, despite the consistent cognitive domains of memory (learning and remembering), attention and executive function that are commonly affected in patients with MDD, MoCA does not have a clear-cut subscale for thorough assessment of executive function except for verbal fluency. Hence, a more comprehensive tool is suggested for future studies” |
|
3 |
One another thing is that (Page 2, lines 87-92 and abstract) It seems you have sample size mistake. while you have calculated your sample size 244, but you included only 80 participants with MDD. |
Thank you very much for the comment. The sample size calculation was accurate based on single proportion for cross sectional study. |
|
4 |
minor mistakes: |
Thank you for the input. We have omitted reference 3 in the revised manuscript.
|
Reviewer 2 Report
This study is important and will guide future studies in the field and clinical practice. However, the authors should improve the manuscript before publishing the study.
I would recommend this article after the authors have made corrections.
Introduction
- In the abstract and the introduction, the authors mention that there are limited tools available to assess cognition in MDD patients. It is not true, there are several tests to evaluate attention, memory, executive functions, etc. I don’t think it is a good argument to justify the study. Furthermore, the MOCA is only a screening tool, I do not recommend focusing on this argument. In fact. The use of the MOCA could even be a limit of the study since future studies should assess cognitive functions with a more exhaustive battery of cognitive tests.
Methods
- Can you name or detail the items of the M.I.N.I A1-A3, and A4?
- Line 126, the MoCA is sensible and specific to what? It is unclear to me.
Results
Discussion
- In the 2nd paragraph, I suggest adding more hypotheses on why attention, memory, and visuospatial/executive functions are more altered. Is there any neurobiological explanation?
- At the end of the third paragraph, this hypothesis of the cognitive reserve could explain why people with more education or less vulnerability to cognitive impairment. https://www.sciencedirect.com/science/article/pii/S0028393209001237
Author Response
|
No |
Comment |
Response |
|
1 |
In the abstract and the introduction, the authors mention that there are limited tools available to assess cognition in MDD patients. It is not true, there are several tests to evaluate attention, memory, executive functions, etc. I don’t think it is a good argument to justify the study. Furthermore, the MOCA is only a screening tool, I do not recommend focusing on this argument. In fact. The use of the MOCA could even be a limit of the study since future studies should assess cognitive functions with a more exhaustive battery of cognitive tests |
Thank you very much for this comment. We now realise that these statements appeared rather confusing. So, it has been omitted accordingly. (Kindly refer to abstract and introduction: page 1 line 13-15 and page 2 line 86-88 respectively.) |
|
2 |
Methods
- Can you name or detail the items of the M.I.N.I A1-A3, and A4?
- Line 126, the MoCA is sensible and specific to what? It is unclear to me.
|
Item A1 explores persistent low mood while item A2 is related to anhedonia. Item A3, explored depressive symptoms according to DSM-5 criteria. For item A4, it denotes significant impairment of functioning level due to depressive symptoms. We have incorporated these statements accordingly in the revised manuscript. (Kindly refer to line 151-154)
Montreal Cognitive Assessment (MoCA) tool, which is a brief cognitive screening test with high sensitivity and specificity to detect cognitive impairment. We have revised the sentence to read as such in the revised manuscript. (Kindly refer to line 160-162) |
|
3 |
Discussion - In the 2nd paragraph, I suggest adding more hypotheses on why attention, memory, and visuospatial/executive functions are more altered. Is there any neurobiological explanation?
- At the end of the third paragraph, this hypothesis of the cognitive reserve could explain why people with more education or less vulnerability to cognitive impairment. https://www.sciencedirect.com/science/article/pii/S0028393209001237
|
Thank you very much for your feedback.
We have added the neurobiological explanation in the 3rd paragraph under discussion to read as “Hippocampus has a vital role in memory. An apparent neuropathological feature of depression is reduction in hippocampal volume which is seen in both MRI (35) and post-mortem (36)studies. High cortisol level in depression is toxic to the hippocampus which leads to its decline (37). Dysregulation of the dorsal and lateral prefrontal cortex while performing executive tasks has been observed in depressed patients in functional neuroimaging study (38)”.
Thank you very much for the input. We have added about the cognitive reserve theory in the 4thparagraph to read as ”In other words, higher education attainment provided better protection against cognitive decline. This observation could possibly be explained by the concept of cognitive reserve. Cognitive reserve refers to the ability to make flexible and efficient use of cognitive networks when performing tasks in the presence of brain pathology (39). Cognitive reserve provides a ready explanation for why many studies have demonstrated that higher levels of intelligence as well as educational and occupational attainment are good predictors of which individuals can sustain greater brain damage before demonstrating functional deficit” (40)
|

Reviewer 3 Report
Thank you for allowing me to review this manuscript, entitled “Cognitive Dysfunction and its Associated Factors in Patients with Major Depressive Disorder”. As implied by the title, the authors have employed Mini International Neuropsychiatric Interview (MINI) to solicit the presence of Major Depressive Disorder (MDD) (n= 245). The cognitive status was measured by bedside cognitive measures known as the Montreal Cognitive Assessment (MOCA). To tease out the associated factors, the authors also collected various socio-demographic variables (age, gender, ethnicity (race), marital status, education level, employment status, household income) and clinical variables. Using MOCA, approximately 33% of the samples exhibited cognitive decline. The factors associated with cognitive decline included education, relapsing subtypes of MDD, treatment non-compliance and medical comorbidity. As most of the cognitive studies on mood disorders have largely emerged from the Euro-Amerian population, this study sentinel type from the global south. Some comments to the authors are detailed below.
CONCEPTUAL ISSUES
The ‘purist’ in neuropsychology will refuse the idea that MOCA is a measure of cognitive impairment. To circumvent this issue, the authors could use the term such as cognitive decline or global cognitive decline. Being bedside, MOCA is only equipped to reveal cognitive decline. It is no wonder this study has received approximately 33% have cognitive decline. This might explain the presently observed spurious rate.
With the spectrum of depressive disorders in existence, it is not clear why the authors used, a brief fully structured interview, the MINI to diagnose the presence of MDD. More protracted semi-interview such as Composite International Diagnostic Interviews for ICD or Structured Clinical Interviews for DSM would have been essential. This should be mentioned as a limitation because a brief instrument such as MINI tends to lead to a false positive.
Could the authors consider the term ethnicity rather than race?
INTRODUCTION
In essence, the study examined the frequency of cognitive decline and associated factors. I think the authors could have a paragraph on the prevalence of cognitive decline in MDD. Also, the authors could introduce literature on the associated factors of cognitive decline in MDD. The literature is abundant on these topics, but still uncharted territory in Malaysia.
METHODS
The method section is well written. However, for brevity, the authors need to have subheadings. I will encourage the authors to structure their method section as defined by the checklist known as STROBE (Strengthening the reporting of observational studies in epidemiology) shown here (https://www.strobe-statement.org/checklists/). For example, these statements (“Approval for this study was obtained from the Research Ethics Committee … Protocol number NMRR-19-702-46518”) could be preceded with a subheading entitled ‘Ethical Approval’
RESULT & DISCUSSION
The description of the result is forthright and succinct
The authors need to acknowledge the consistency of cognitive profile of people with MDD, including executive function, learning and remembering (memory) as well as attention. While MOCA has a subscales tapping into memory and attentional capacity, there is not a clear cut test that comes under executive functioning except remotely verbal fluency.
LIMITATION
This statement is not clear (“Secondly, current severity of depression was not addressed that could have an impact on cognitive function”)
The bulk of the participants was in the age group > 46. If the existing science has heuristic value, then this age group is likely to be marked with subtle cognitive decline or age-related cognitive decline. This should be highlighted as a limitation.
REFERENCE
The authors employed 26 references in their study. Most of them are recent and relevant.
Author Response
|
No |
Comment |
Response |
|
1 |
CONCEPTUAL ISSUES
The ‘purist’ in neuropsychology will refuse the idea that MOCA is a measure of cognitive impairment. To circumvent this issue, the authors could use the term such as cognitive decline or global cognitive decline. Being bedside, MOCA is only equipped to reveal cognitive decline. It is no wonder this study has received approximately 33% have cognitive decline. This might explain the presently observed spurious rate |
Thank you very much for this feedback.
We agree with the statement. Hence, we have revised the manuscript (entirely) to replace “dysfunction” with “decline”. Kindly refer the revised manuscript in which the changes have been highlighted.
|
|
2 |
With the spectrum of depressive disorders in existence, it is not clear why the authors used, a brief fully structured interview, the MINI to diagnose the presence of MDD. More protracted semi-interview such as Composite International Diagnostic Interviews for ICD or Structured Clinical Interviews for DSM would have been essential. This should be mentioned as a limitation because a brief instrument such as MINI tends to lead to a false positive. |
Thank you for pointing this out. This has been duly amended and highlighted in the revised manuscript as a 4th limitation
|
|
3 |
Could the authors consider the term ethnicity rather than race. |
This has been aptly corrected. |
|
4. |
INTRODUCTION
I think the authors could have a paragraph on the prevalence of cognitive decline in MDD. Also, the authors could introduce literature on the associated factors of cognitive decline in MDD. The literature is abundant on these topics, but still uncharted territory in Malaysia.
|
Thank you very much for the input. We have added two paragraphs (under introduction section) on prevalence of cognitive decline including data from Malaysia in MDD (3rd paragraph) and its associated factors (4th paragraph).
|
|
5. |
METHODS
However, for brevity, the authors need to have subheadings. I will encourage the authors to structure their method section as defined by the checklist known as STROBE (Strengthening the reporting of observational studies in epidemiology) shown here (https://www.strobe-statement.org/checklists/). For example, these statements (“Approval for this study was obtained from the Research Ethics Committee … Protocol number NMRR-19-702-46518”) could be preceded with a subheading entitled ‘Ethical Approval’
|
Thank you very much for the response. We have amended the methods section to follow STROBE checklist under subheadings of Study Population, Study assessments, Ethical Approval and Statistical Analyses.
|
|
6. |
RESULT & DISCUSSION
The description of the result is forthright and succinct
The authors need to acknowledge the consistency of cognitive profile of people with MDD, including executive function, learning and remembering (memory) as well as attention. While MOCA has a subscales tapping into memory and attentional capacity, there is not a clear cut test that comes under executive functioning except remotely verbal fluency.
|
Thank you for the comments.
Thank you very much for the reply.
We have acknowledged this as one of the limitations for our study under the discussion section.
|
|
7. |
Under limitation. This statement is not clear (“Secondly, current severity of depression was not addressed that could have an impact on cognitive function”) |
This ambiguous statement has been duly removed under discussion (limitation) |
|
8 |
The bulk of the participants was in the age group > 46. If the existing science has heuristic value, then this age group is likely to be marked with subtle cognitive decline or age-related cognitive decline. This should be highlighted as a limitation |
Thank you for pointing this out. This has been modified and highlighted in the revised manuscript as the 2nd limitation to replace the existing ambiguous limitation. |
